# Can Better Solvers Find Better Matches? Assessing Math-LLM Models in Similar Problem Identification

**Savitha Sam Abraham**[*]**, Pietro Totis**[‡]**, Marjan Alirezaie**[†] **, Luc De Raedt**[§,‡]

[*]Australian Institute for Machine Learning, The University of Adelaide, Australia
savitha.samabraham@adelaide.edu.au
[†]Flybits Labs. TMU Creative AI Hub, Toronto, Canada
marjan.alirezaie@flybits.com
[§]Örebro University, Centre for Applied Autonomous Sensor Systems(AASS), Örebro, Sweden
luc.de-raedt@oru.se
[‡] Department of Computer Science, KULeuven, Belgium
pietro.totis@outlook.it, luc.deraedt@kuleuven.be

## Abstract

Researchers have adapted large language models (LLMs) for mathematical reasoning by fine-tuning them with math-specific datasets to create math-specialized LLMs. This paper evaluates such models not only on solving accuracy but also on their ability to identify similar problems. We introduce an indicator task—retrieving a similar problem given a query word problem—to assess whether the model's internal representations of the word problems capture mathematical semantics. A model capable of solving a problem should also be adept at identifying problems requiring similar reasoning, as human experts do. Using a dataset of Probability Word Problems with formal symbolic annotations, we show that math-specialized LLMs often prioritize linguistic similarity over mathematical similarity. This underscores the need for symbolic intermediate representation during fine-tuning of a LLM to better capture mathematical essence of a problem aiding improvement in model's consistency and reliability.

## Introduction

Since their introduction, large language models (LLMs) have been applied to mathematical reasoning, as proficiency in mathematical tasks indicates strong cognitive skills (Iuculano and Menon 2018). A math word problem is a mathematical task expressed in natural language. Solving it requires linguistic skills to understand the problem and translate it into its mathematical form, as well as mathematical skills to solve the task embedded within the problem. Researchers have explored the application of LLMs to math word problem solving, given the vast amount of data they are trained on, which enables them to possess advanced linguistic skills and embedded mathematical and world knowledge. While significant progress has been made, gaps remain, primarily due to the brittleness of LLMs in handling word problems in general (Ahn et al. 2024). This has prompted rigorous efforts from researchers to enhance the performance of LLMs in mathematical reasoning. A common approach among research groups is to fine-tune LLMs, already pre-trained on large generic datasets, for mathematical tasks

using data specifically designed for mathematical reasoning (Wang et al. 2023), (Imani, Du, and Shrivastava 2023). Consequently, research is increasingly focused on acquiring large-scale training data in the form of (*math word problem, solution with an explanation*) pairs (Toshniwal et al. 2024).

In this paper, we analyze several recent math-LLMs, including Qwen (Yang et al. 2024), Mathstral (a math model from Mistral) (Jiang et al. 2023), and DeepseekMath (Shao et al. 2024). Our objective is to examine whether additional fine-tuning on math-specialized data truly enhances the models' understanding of the domain's semantics. We quantify this semantic understanding by evaluating the performance of these models on an auxiliary indicator task: given a query math word problem and a knowledge base (KB) of word problems, retrieve the problem from the KB that is most similar to the query. We consider this retrieval task an ideal indicator task to assess whether the internal representation of a problem used by the math-LLM effectively captures the problem's semantics with respect to its mathematical essence. The concept of indicator tasks for evaluating dense vectorial representations has been extensively studied (Levy, Goldman, and Tsarfaty 2023).

We focus on a specific subset of mathematical reasoning problems, namely, probability word problems. Dries et al. introduced a dataset, $NLP4PLP$, of probability word problems where each problem was annotated with its solution and a formal representation based on a declarative programming language (Dries et al. 2017). Problems with similar formal representations share similar solving strategies, meaning they are mathematically alike. The availability of such formal representations enables a systematic evaluation of math-LLMs not only in terms of accuracy but also in terms of consistency—i.e., whether similar problems are treated similarly—and the semantics captured by the internal representation of a problem, as measured through the indicator retrieval task described.

The paper is organized as follows - the dataset is described in detail, following which we present the experiments performed along with an analysis of the results.

## Dataset

Dries et al. (Dries et al. 2017) introduced a dataset of probability word problems collected from online sources and textbooks. The dataset contains 2160 problems manually annotated by three job students. Here is an example problem from the dataset:

> **Problem P1**: A class of 20 boys and 10 girls has 15 Math majors and 15 Physics students. What is the probability of picking a student who is both a boy and a physics major?

The annotation encompasses the solution and a formal representation based on the declarative language introduced in the paper. The language defines a set of predicates that are instantiated with arguments to describe the objects, their arrangement, and the questions of the problem. Objects are described by means of the predicates `group`, `size`, `property`, and `given`. The first two statements' arguments define, respectively, the name of a set of objects and its cardinality. The `property` statement declares the relevant attributes of the objects. Finally, the `given` predicate expresses the number and combinations of observable properties, taking as argument a constraint. Constraints quantify objects or properties in a group, using dedicated predicates (`exactly`, `at_least`, `some`, `all`,...). Boolean predicates (`and`, `or`, `not`) are available to form complex constraints. For example, the following statements describe the set of objects of Problem 1:

```
group(students).
property(gender, [boy, girl]).
property(major, [physics, math]).
given(exactly(10, students, girl)).
given(exactly(20, students, boy)).
given(exactly(15, students, physics)).
given(exactly(15, students, math)).
```

The statements `take` and `take_wr` define actions on the initial set that produce a new arrangement, its specific form can be prescribed with the predicate `observe`. The statements `take` and `take_wr` describe the probabilistic selection of a given number of objects from the initial set. `take_wr` denotes that an object can be selected multiple times (selection *with* replacement), conversely, `take` denotes that each object can be selected at most once (selection *without* replacement). For example the action "picking a student" is defined by the statement:

```
take(students, pick, 1).
```

`observe` is similar to `given`, but while `given` describes the inital set of objects, `observe` describes the result of a `take` or `take_wr` action.
Finally, a `probability` predicate defines a question of the problem, taking as argument a constraint: the goal is to find the probability that the final arrangement satisfies the given constraint. For example the probability of picking "a boy and a physics major" is declared as:

```
probability(and(all(pick, boy),
all(pick, physics))).
```

It is worth noting two aspects of this declarative language: first, the statements are order-invariant, second, they define

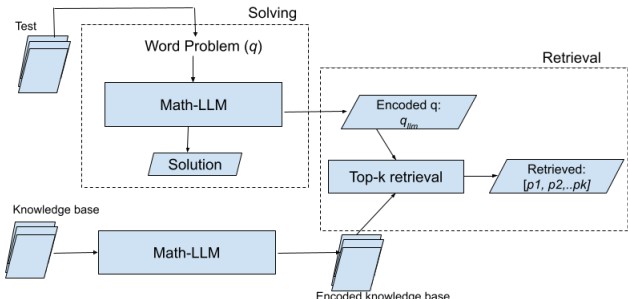

Figure 1: Experiments - given the world problem $q$, (1) generate the solution (2) retrieve the top-k most similar problems $p_1, ..., p_k$ from the knowledge base.

only *what* the problem is about, and not *how* to compute the solution. How to compute the solution is devolved to a dedicated solver based on the probabilistic logic programming tool ProbLog (Fierens et al. 2015). Therefore this dataset is designed for an end-to-end approach based on two steps: first the translation of the problem from natural language to its formal representation; second, the application of symbolic reasoning techniques to solve the formal definition of the problem. For the experiments in this paper, we leverage the formal representations of the problems to identify similar problems in the dataset - problems with similar formal representations are likely to adopt similar solving strategies. Further details about the language and solver can be found on the project's webpage [1].

## Experiments

The objectives of our experiments are to evaluate the performance of recent Math LLM models, focusing on their ability to:

- **Solve problems accurately (Accuracy)**: The Math-LLM model is used to solve each problem $q$ in the test set.

- **Treat similar problems similarly (Consistency)**: Here, we analyse the inconsistencies in the behavior of a Math-LLM model. If $q_1$ and $q_2$ are two problems with similar mathematical forms or solving strategies, we estimate the inconsistency in the Math-LLM model by counting the number of instances where the Math-LLM model solves only one of them accurately.

- **Represent problems while capturing their mathematical semantics (Semantic Representation Quality)**: We assess the quality of a Math-LLM model's internal representation of a word problem by evaluating the task of similar problem identification. For each problem $q$ in the test set, the goal is to retrieve the problem from a specified knowledge base (KB) of problems that have the same mathematical form. Notably, there is exactly one such problem in the KB, denoted as $q_{similar}$, that shares the same mathematical archetype as the query $q$. The objective is to evaluate whether $q_{similar}$ is included in the top-$k$

---

[1]https://dtai.cs.kuleuven.be/problog/natural\_language/

retrieved results from the KB. We use the FAISS (Facebook AI Similarity Search) library (Douze et al. 2024) for efficient similarity search.

Figure 1 illustrates the experiments performed to evaluate the different Math-LLM models with respect to the three objectives mentioned above.

## Models

In this paper, we compare three of the Math-LLM models released this year - February 2024 - DeepSeekMath, July 2024 - Mathstral, and August 2024 - Qwen2.5-Math.

- **Qwen 2.5-Math-Instruct-7B** (Yang et al. 2024) is a model built on top of its precursor model, Qwen 2-Math. Pre-training and post-training of Qwen 2.5-Math model (supervised fine-tuning and reward model) include high-quality mathematical data generated by Qwen 2-Math model which optimizes the model's performance. Qwen 2.5 model also used the trained reward model during the inference stage to guide sampling.

- **DeepSeekMath** (Shao et al. 2024) is a Math LLM model built from the predecessor model, DeepSeek-Coder-Base-v1.5 by continuing to pretrain it with publicly available mathematical data from Common Crawl and code data.

- **Mathstral** is yet another Math LLM model, and it is built on top of Mistral (Jiang et al. 2023).

## Data: Test Set and Knowledge Base

**Knowledge Base (KB)**: Utilizing the formal representations of the problems described in Section Dataset, we construct a KB of probability word problems. This KB is designed to encompass the entire dataset in terms of the mathematical form of the problems. Each ***mathematical archetype*** is characterized by the predicates present in its formal representation.

The predicates in the intermediate declarative programming language used to represent the intermediate programs are: *aggcmp, observe, nth, atleast, atmost, all_same, all_diff, more_than, take_wr, size, none, max, min, less_than, not, and, or, all, union, some, sum, is_even, is_odd, =:=, >=, >, <, rest, outcome*. A specific combination of these predicates defines a mathematical archetype. For instance, problems whose formal representations include logical operators such as *and* and *or* define one archetype, while problems that include these logical operators alongside comparison operators like *max* and *min* define another archetype, and so on. We identified 282 distinct mathematical archetypes, and the KB contains a total of 282 problems, each corresponding to a unique mathematical archetype.

**Test Set**: The test set is created by randomly selecting 375 questions from the remaining dataset.

**Similar problems**: Similar pairs are formed by matching each problem in the test set with the problem in the KB that shares the same mathematical archetype (notably there is exactly one such problem in the KB).

## Evaluation

We evaluate the three models above using the following metrics:

- **Accuracy**: The accuracy measures the percentage of problems the model answers correctly.

- **Inconsistency**: To assess the consistency of a model as a solver, we examine if the model handles similar problems in the same way. These problems may differ linguistically but share the same mathematical archetype. Inconsistency is measured as the percentage of similar pairs where the model solves only one accurately.

- **Recall@k** For each problem in the test set, we retrieve the most similar problem from KB. The standard metric Recall@k measures the retrieval performance (Harman 2011).

$$\text{Recall@k} = \frac{\text{No. of queries where } q_{similar} \text{ is in top k retrieved}}{\text{Total number of queries}} \tag{1}$$

## Results

Table 1 presents the results—accuracy, inconsistency, and Recall@k—for the three Math-LLM models. Among them, the Qwen2.5 Math model demonstrates the highest accuracy. This can be attributed to its training pipeline, which leveraged high-quality mathematical data generated by its predecessor, the Qwen2 Math model, during supervised fine-tuning and reinforcement learning stages.

Notably, higher solving accuracy is accompanied by reduced inconsistency in the model's outputs. In other words, problems with similar archetypes are solved more consistently by the models, and vice versa.

However, a significant limitation shared by all three models is their inability to effectively capture the mathematical semantics of problems within their internal representations. When word problems are represented using Qwen2.5 Math-based embeddings, closer embeddings often indicate linguistic rather than mathematical similarity. As shown in Figure 2, a query word problem set in the context of "color and balls" is retrieved as more similar to another problem with the same context. The problem sharing its mathematical semantics does not appear among the top 10 retrieved results as it is set in a completely different context of "store and magazines".

It is also worth highlighting that retrieval performance does not correlate directly with solving accuracy. For instance, while DeepSeekMath achieves better accuracy than Mathstral, the latter outperforms it in retrieval tasks.

## Discussion

The evaluation of the three Math-LLM models—Qwen2.5 Math, DeepSeekMath, and Mathstral—provides valuable insights into their problem-solving capabilities and internal representations of mathematical concepts. Incorporating higher-quality mathematical data during training, as done in Qwen2.5, enhances the solving accuracy of Math-LLM models. This raises an important research question: How can we further enhance the quality of mathematical data used for

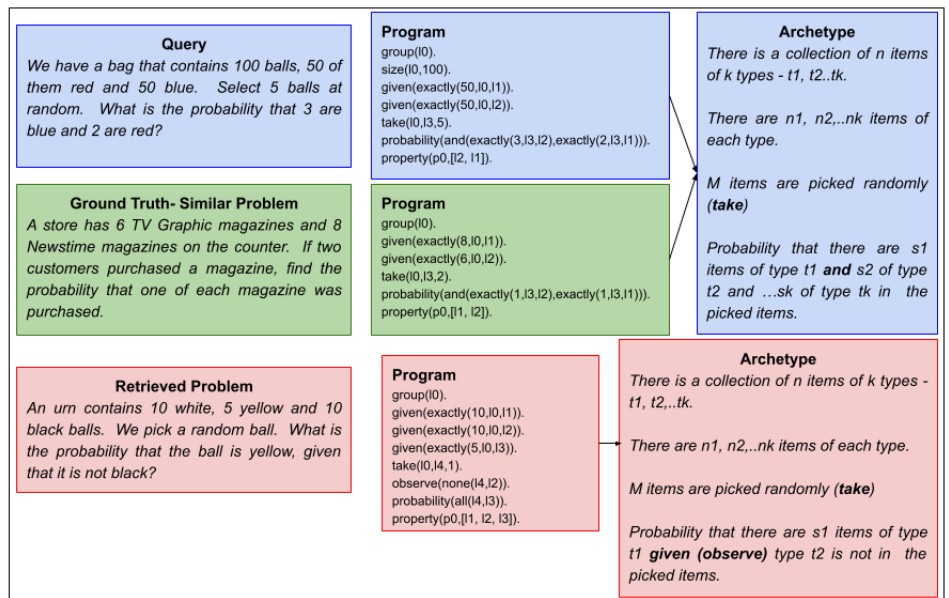

Figure 2: An example: Comparison of similar problem retrieved using Qwen2.5 embeddings with the ground truth similar problem in KB.

| Model | Accuracy (%) | Inconsistency (%) | Recall@10 (%) |
|---|---|---|---|
| Qwen2.5-Math | 85.86 | 24.92 | 12.26 |
| DeepSeekMath | 61.86 | 43.08 | 4.53 |
| Mathstral | 53.6 | 52.20 | 12.53 |

Table 1: Accuracy, Inconsistency, Semantic Representation Quality using $Recall@k$, with $k = 10$ of Qwen 2.5-Math, DeepSeek-Math and Mathstral models.

training such large models while simultaneously reducing the amount of data required?

Despite the improvement in solving accuracy achieved, all three models share a common limitation: an inability to adequately capture the mathematical semantics of problems within their internal representations. This indicates that the models are more attuned to surface-level language features than to the deeper mathematical relationships that define problem archetypes.

These findings underscore the need for enhancements in how Math-LLMs internalize and represent mathematical semantics. Improving the semantic representations could lead to models that not only solve problems more accurately but also retrieve and relate problems based on their mathematical structures rather than merely their linguistic features. This would be particularly beneficial for applications that require understanding and manipulating mathematical concepts, such as educational tools that provide personalized learning experiences.

Future research should focus on developing techniques that enable models to better grasp the mathematical semantics inherent in problems. This could involve augmenting training datasets with intermediate mathematical representations of the problems that emphasizes mathematical relationships. Incorporating data augmented with such semantic structures into the training process of these models would encourage them to learn patterns that map mathematical archetypes to solving strategies, instead of relying on just the linguistic features. Notably, relying on linguistic features can also increase the data requirements, as similar linguistic patterns may correspond to entirely different solution strategies. By shifting the focus to mathematical semantics, we could reduce the reliance on large datasets.

## Conclusion

This study evaluates the performance of three recent Math-LLM models highlighting their strengths and limitations in solving mathematical problems and capturing mathematical semantics. While incorporating higher-quality mathematical data during training, as seen with Qwen2.5 Math, improves solving accuracy, all models still rely heavily on linguistic features rather than mathematical semantics in their internal representations. This limitation impacts their ability to retrieve and relate problems based on underlying mathematical structures. To address these challenges, in future we would like to explore methods to integrate semantically rich data into training, emphasizing mathematical relationships over linguistic patterns.

## Acknowledgements

This research was financially supported by the Centre for Augmented Reasoning (CAR), an initiative by the Department of Education, Australian Government and Wallenberg AI, Autonomous Systems, and Software Program (WASP), Sweden.

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
