# OpenReview forum: "Can Better Solvers Find Better Matches? Assessing Math-LLM Models in Similar Problem Identification"
_AAAI.org/2025/Workshop/NeurMAD — AAAI 2025 Workshop NeurMAD Submission_

### Official Review · Reviewer_DkBg · 2024-12-20
**Good paper, missing some details**

**Rating:** 7
**Confidence:** 3

**Review:**

### Summary
This paper proposes a new evaluation strategy for models specialized in mathematical reasoning. Rather than simply evaluating the accuracy of these models when solving these tasks, the authors propose to complementarily test their ability to identify similar problems. The rationale behind this is that a true understanding of a mathematical problem implies the construction of a proper abstract model for it, which should come with the ability to evaluate similarities between these constructed models. To test this, the authors use NLP4PLP, a dataset of probability word problems where each problem is annotated with its solution and a formal representation based on a declarative programming language. Three different Math-LLMs are evaluated: Qwen, DeepSeekMath, and Mathstral. In the experiments, the authors report three different metrics: accuracy (% of problems solved), inconsistency (% of pairs of problems that share the same archetype but who are not solved in the same way), and recall@10 (% of problems where the correct matching problem is retrieved from the KB). The experimental results show that, while math-LLMs can solve NLP4PLP problems with good accuracy, their internal representations predominantly encode linguistic rather than mathematical similarity.

### Review
I liked the idea proposed in this paper: simple, clear, and well-documented. To the best of my knowledge, probing the mathematical semantic similarity between internal representations of different models is a novel idea. The closest work is probably [1], where the authors tested the ability of general (and not math-tuned) models to generate (linguistically and not mathematically) similar problems.

I have, however, a few concerns/observations regarding the experiments:

- For the inconsistency metric, it is not clear how the pairs are defined. Do you group all the problems sharing the same archetypes in pairs of two, or do you create a full cartesian product of the problem instances? It would be helpful to clarify this in the manuscript.

- Always on inconsistency, I think it would be good to evaluate the impact of pairs where both examples could not be solved.

- It is not clear how the problem embeddings are extracted. This could have a great impact on the evaluation: intuitively, earlier layers might contain more linguistic information, while final layers could contain more mathematical information. Having some ablation on this, then, could also be interesting.

On a more general note, I think these results are not surprising. Math-LLMs, despite being fine-tuned for mathematical reasoning, are still language models. Hence, the fact that substantial linguistic information is still contained in their “problem models” is expected. I think that, in future work, it would be interesting to understand to what extent it is possible to separate mathematical and linguistic information in the embeddings.

[1] Solving Math Word Problems concerning Systems of Equations with GPT-3, Zong, and Krishnamachari

---

### Official Review · Reviewer_penM · 2024-12-20
**Do LLMs truly capture mathematical semantics**

**Rating:** 6
**Confidence:** 3

**Review:**

## Summary
The paper addresses an important field of LLM evaluations.  Its focus on semantic understanding over pure accuracy is novel and positions it as a valuable contribution to mathematical reasoning and AI explainability. The authors use the NLP4PLP, a dataset of probability words, and evaluate three different LLMs: Qwen, DeepSeekMath, and Mathstral.

## Strengths

**Novel Evaluation Approach:** The evaluation approach present in the paper focuses on semantic and similar problem understanding rather than just accuracy

**Valuable Results:**  The authors show that Math tuned models still rely heavily on linguistic features rather than mathematical semantics in their internal representations

## Weaknesses

**Limited Evaluation:** The authors only evaluate their results on a small dataset comprising probability-related problems. It would be interesting to see if these results are similar to those of other fields of mathematics, such as algebra.

**Lack of reasoning behind the observed results:** While the authors demonstrate models' reliance on linguistic features over mathematical semantics in their representations, they lack theoretical analysis or hypotheses explaining this phenomenon in depth. Adding the same in more detail would strengthen the paper's contributions.

---

### Official Review · Reviewer_XMK6 · 2024-12-29
**Good research question, but the paper could be further enhanced**

**Rating:** 5
**Confidence:** 4

**Review:**

## Summary
This paper examines whether Math-LLMs are able to perform consistently on different math problems with similar solutions. The research is conducted on three well-known math-LLMs: Qwen-Math, Mathstral, and DeepSeek-Math. The testbed is a set of probability word problems. These problems are annotated with formal representations. The author conduct experiments to explore whether Math-LLMs could treat similar problems similarly and capture mathematical semantics in these problems.

## Pros
- Regarding the research question. It is known that LLMs are black-box models. This paper provides a new way to uncover the inner mechanism of LLMs by questioning whether they behave consistently on semantically similar problems.

## Cons
- The organization of this paper could be further improved.
    1. Typo: in subsection{Models}: Qwe2.5-Math --> Qwen2.5-Math
    2. The experiment procedure is only depicted in Figure 1. However, the reviewer believe it is important to elaborate on the details of the conduction.
    3. The authors should further explain why we can use the embedding similarity to measure semantic similarity.

The reviewer would support its acceptance to the workshop if the author could properly address Cons 2 and Cons 3

---

### Decision · Program_Chairs · 2024-12-30

**Decision:**

Accept

**Comment:**

We agree with the major opinions of the reviewers, and this paper fits into one of the focuses of this workshop.